# Synthesis, Properties, and Application of Small-Molecule Hole-Transporting Materials Based on Acetylene-Linked Thiophene Core

**DOI:** 10.3390/molecules28093739

**Published:** 2023-04-26

**Authors:** Hui-Juan Yu, Jing Xiao, Jian Chen, Xuefeng Ren, Ya-E Qi, Xuemei Min, Guang Shao

**Affiliations:** 1Key Laboratory of Hexi Corridor Resources Utilization of Gansu Universities, College of Chemistry and Chemical Engineering, Hexi University, Zhangye 734000, China; 2Guangdong Key Laboratory of Animal Conservation and Resource Utilization, Guangdong Public Laboratory of Wild Animal Conservation and Utilization, Institute of Zoology, Guangdong Academy of Sciences, Guangzhou 510260, China; 3School of Chemistry, Sun Yat-sen University, Guangzhou 510006, China; 4Shenzhen Research Institute, Sun Yat-sen University, Shenzhen 518057, China

**Keywords:** perovskite solar cells, hole-transporting materials, thiophene, acetylene, intramolecular charge transfer

## Abstract

Three small molecule organic compounds based on conjugated acetylene-linked methoxy triphenylamine terminal groups with different substituted thiophene cores were synthesized and firstly applied as hole-transporting materials (HTMs). The electron-deficient acetylene linkers can tune the energy levels of frontier molecular orbitals. The physical property measurements show that the HTMs (**CJ-05**, **CJ-06,** and **CJ-07**) possess good stability, hydrophobicity, and film-forming ability. Further, the HTMs were applied in the MAPbI_3_-based perovskite solar cells (PSCs), and the best power conversion efficiency (PCE) of 6.04%, 6.77%, and 6.48% was achieved, respectively, which implies that they exhibit great potential in photovoltaic applications.

## 1. Introduction

The solution-processable organic–inorganic hybrid perovskite solar cells (PSCs) are among the most promising photovoltaic technologies because of the rapid enhancement of the power conversion efficiency (PCE) from 3.8% [1] in 2009 to 25.7% [2] as recently certified, which is comparable to the commercialized silicon cells. However, the industrialization of PSCs remains unfulfilled due to the poor long-term stability of perovskite materials under circumstances including moisture, oxygen, heat, and illumination [3,4,5]. Numerous research works have proved that the modification of interfaces and the optimization of charge-transporting materials are effective routes to enhance the performance and stability of PSCs [6,7,8]. The most efficient and frequently applied PSCs contain a conventional negative-intrinsic-positive (n-i-p) structure, in which a hole-transporting layer is deposited on the top of perovskite active layer, thus protecting it from the outer moisture and oxygen [9]. Therefore, an ideal HTM should possess multiple properties, including matched frontier molecular orbital levels, high hole mobility, fine film formation, good stability and solubility, and low-cost synthesis [10,11,12,13,14,15,16]. However, there is not an HTM that can meet all the above requirements. For example, 2,2′,7,7′-tetrakis(*N*,*N*-di-*p*-methoxyphenylamine)-9,9′-spirobifluorene (spiro-OMeTAD [17,18]), known as one of the most efficient small molecule HTMs and, thus, most commonly used as a benchmark for the assessment of other developed HTMs, suffers from low intrinsic hole mobility, dependence on hydrophilic dopants, and high preparation costs. Thus, the research of ideal HTMs is still an unaccomplished challenge.

The thiophene derivates are very commonly utilized building blocks in various organic semiconductors because of their excellent chemical stability, easy modification, and electron richness that facilitates charge transport [19,20,21]. Particularly in the case of HTMs for PSCs, the thiophene-containing compounds are expected to promote the charge extraction and defect passivation in the perovskite layer via probable S-Pb interaction [22]. A simple HTM based on triphenylamine groups and a thiophene core has been reported by several research groups, and the PCEs up to 15% were acquired [23,24,25,26,27,28,29]. 3,4-Ethylenedioxythiophene (EDOT) moiety, which is widely used in numerous organic electronic materials, has also been chosen as the central core for the HTM, and a moderate PCE of almost 14% was obtained [30]. Recently, our group has developed an HTM based on 3,4-phenylenedioxythiophene (PheDOT), a more planar and stable core, to promote molecular packing and interaction; thus, the hole mobility was greatly improved. The HTM was successfully implemented to the MAPbI_3_-based PSCs [31]. Therefore, some ingenious modifications for the HTMs based on thiophene derivatives could be expected to further improve their photovoltaic performances.

On the other hand, the acetylene group has drawn our attention as a promising motif for the modification of HTMs because of their extensive applications in conductive polymers. In recent years, it has also been applied to the design of HTMs for PSCs [32,33,34]. The acetylene group possesses a rigid linear structure and electron-withdrawing behaviour, which are supposed to regulate molecular aggregation and adjust molecular energy levels. Moreover, the connection of electron-acceptor (A) acetylene groups with electron-donor (D) triphenylamine/thiophene groups could form a D-A-D alternating structure. Thus, an intramolecular charge transportation between donor and acceptor units during excitation could significantly enhance the dipole moments of the HTM molecule, creating a built-in potential that promotes hole extraction [35,36,37]. Our group has previously reported an acetylene-linked HTM based on a 9,9′-bicarbazole core and received a best PCE of approximately 16.08% in triple cation PSCs [38]. These results indicate that more research efforts should be given to the acetylene introduction strategy for the creation of efficient HTMs.

On the basis of the reported HTMs, we have designed and synthesized three small molecule HTMs (**CJ-05**, **CJ-06,** and **CJ-07**), in which a pair of acetylene linkers were constructed between different substituted thiophene cores and terminal triphenylamine groups. Although the compounds **CJ-05** and **CJ-06** have been reported in the literature [39], they have not yet been applied as HTMs. The purpose of the design is to modulate the frontier molecular orbitals and enhance charge transfer. A series of physical tests, including thermal stability, optical spectra, electrochemical properties, hole mobility, hydrophobicity, and film morphology, were carried out for these compounds to investigate their capability and quality as HTMs. Finally, the HTMs were applied into n-i-p PSCs, and the photovoltaic performances were measured. The introduction of acetylene linkers was expected to be an effective strategy to generate excellent HTMs that could promote the commercialization of PSCs.

## 2. Results and Discussion

### 2.1. Syntheses

**CJ-05**, **CJ-06,** and **CJ-07** were synthesized via Sonogashira coupling between ethynyl substituted triphenylamine and corresponding 2,5-dihalogenated thiophenes (see Figure 1, more detailed synthetic route is illustrated in Appendix A). The construction of ethynyl triphenylamine moieties originated from the copper-catalyzed C-N coupling of aniline and 4-iodoanisole, followed by electrophilic iodination, Sonogashira coupling with trimethylsilylacetylene, and deprotection successively to introduce the ethynyl group. On the other hand, the thiophene core precursor is either directly purchased (2,5-dibromothiophene for **CJ-05**) or synthesized from transetherification of commercially available 3,4-dimethoxylthiophene with glycol (**CJ-06**) or catechol (**CJ-07**), then treated with electrophilic iodination. The chemical structures of all synthesized intermediates and HTMs have been confirmed by nuclear magnetic resonance and high-resolution mass spectroscopy (Appendix A). The three HTMs are all highly soluble in common organic solvents, such as dichloromethane, tetrahydrofuran, and chlorobenzene, which makes them capable of solution processing during the fabrication of PSCs.

### 2.2. Thermal Stability

In order to verify whether the synthesized HTMs could endure the severe operating temperature of PSCs, their thermal stability was investigated by thermal gravimetric analysis (TGA) and differential scanning calorimetry (DSC). As illustrated in Table 1 and Appendix A, in the TGA test, **CJ-05** and **CJ-07** showed decomposition temperatures (*T*_d_) corresponding to 5% weight loss of 426.8 °C and 418.8 °C, respectively, which are slightly higher than that of spiro-OMeTAD (417.0 °C [40]), while **CJ-06** showed a significantly lower *T*_d_ of 365.0 °C. Meanwhile, in the first heating of DSC experiment, **CJ-05**, **CJ-06**, and **CJ-07** showed melting points (*T*_m_) of 85.0 °C, 103.3 °C, and 240.6 °C, respectively, while in the second heating, they exhibited glass transition temperature (*T*_g_) of 76.7 °C, 93.5 °C, and 103.6 °C, respectively, which are all lower than spiro-OMeTAD (126.0 °C [40]). Obviously, both *T*_m_ and *T*_g_ of the three HTMs have the same tendency as their molecular weights, and it is also the reason for their *T*_g_ being lower than spiro-OMeTAD. It can be concluded that **CJ-06** and **CJ-07** could meet the operation condition of PSCs (about 80 °C), while **CJ-05** may not be so satisfying.

### 2.3. Optical Property

The optical properties of **CJ-05**, **CJ-06**, and **CJ-07** were studied by UV–Vis absorption and photoluminescence (PL) spectra of their dichloromethane solution (Figure 1a, Appendix A). All three HTMs exhibited two absorption peaks similar to that of spiro-OMeTAD, which includes a minor peak in the range of 260–330 nm and a major peak at 330–500 nm. The former absorption peak could be attributed to the π–π* transition of peripheral triphenylamine units, while the latter one may originate from the π–π* transition of the entire HTM molecule [42,43]. Combined the with the PL emission spectra, the Stokes shifts of **CJ-05**, **CJ-06**, and **CJ-07** were calculated to be 120, 107, and 133 nm, respectively, which are all larger than that of spiro-OMeTAD (41 nm). The larger stokes shifts of the HTMs imply a greater structural deformation during excitation, which indicates better pore filling and more efficient hole extraction in PSCs [44]. The deformation of the three HTMs may originate from the rotatable terminal triphenylamine groups. Additionally, the absorption and emission spectra of each HTM were normalized to provide an intersection wavelength (λ_inter_); thus, the band gaps (*E*_g_) of **CJ-05**, **CJ-06**, and **CJ-07** were calculated to be 2.72 eV, 2.72 eV, and 2.68 eV, respectively, from the formula *E*_g_ = 1240/λ_inter_.

We further examined the UV–Vis absorption spectra of HTM films spin-coated on TiO_2_/MAPbI_3_ substrates. As shown in Figure 1b, all samples displayed almost identical absorption curves covering a wide wavelength range up to near-infrared region (the edge is at approximately 780 nm, which is related to the band gap of MAPbI_3_), which reveals that the perovskite layer contributed the vast proportion to the absorption of all samples. Due to the superimposed characteristic absorption of each component, the four HTMs have a slightly enhanced absorption band between 430 and 480 nm.

### 2.4. Energy Levels

Cyclic voltammetry (CV) measurement of **CJ-05**, **CJ-06**, **CJ-07,** and spiro-OMeTAD in dichloromethane solutions was carried out in order to seek the energy levels of frontier molecular orbitals, and the resulting current–voltage curves are illustrated in Figure 2a. All three HTMs showed highly similar CV curves, with one oxidation and one reduction peak within the scan range of −0.8~0.2 V, and they are all highly reversible, which proves that these HTMs have fine electrochemical stability. The arithmetic mean of the oxidation and reduction potential (*E*_1/2_ = (*E*_ox_ + *E*_red_)/2) for each HTM was extracted from the CV curve, and the energy level of highest occupied molecular orbital (*E*_HOMO_) was then calculated by the equation *E*_HOMO (HTM)_ = *E*_HOMO (spiro-OMeTAD)_ − *E*_1/2 (HTM)_ + *E*_1/2 (spiro-OMeTAD)_, in which the value of *E*_HOMO (spiro-OMeTAD)_ was taken as −5.22 eV according to the reference [41]. The *E*_HOMO_ of **CJ-05**, **CJ-06**, and **CJ-07** turned out to be −5.51, −5.50, and −5.52 eV (Figure 2b), respectively, which are all lower than the valence band energy level of MAPbI_3_ (−5.43 eV, according to the reference [45]). Since the doping process usually lowers the HTM energy level [30], the mismatch of energy levels between perovskite and HTMs will be further expanded if the HTM is doped during device fabrication. The mismatch is unfavorable for hole extraction in the PSCs; thus, a poor photovoltaic performance is predictable. Additionally, the energy levels of the lowest unoccupied molecular orbitals (*E*_LUMO_) of **CJ-05**, **CJ-06**, and **CJ-07** were calculated to be −2.79, −2.78, and −2.84 eV, respectively, from the formula *E*_LUMO_ = *E*_HOMO_ + *E*_g_, where *E*_g_ was obtained from the optical characterization in the previous section. The *E*_LUMO_ of all HTMs are higher than the conductive band of MAPbI_3_ (−3.88 eV), making them capable of blocking the photoexcited electrons. According to the references [29,30,31], the frontier molecular orbital energy levels of the three HTMs are significantly lower compared with the corresponding HTM without the acetylene linkers, proving that the presence of electron-withdrawing acetylene groups has lowered the frontier molecular orbital energy levels.

### 2.5. DFT Simulation

The electronic properties and optimal molecular configuration of the ground state of each HTM were simulated on the Gaussian 09W program. The calculation was carried out under the framework of density functional theory (DFT), with the exchange correlation function of B3LYP and the basis of 6–31 G. The same simulation was also carried out for spiro-OMeTAD, and a result similar to the reference was obtained to verify the simulation reliability [46]. As shown in Appendix A, the HOMOs of **CJ-05**, **CJ-06**, and **CJ-07** are delocalized to the entire molecule, while the LUMOs are localized between the N atoms and the thiophene core. It is noteworthy that the aromatic benzene ring in the PheDOT core of **CJ-07** does not participate in the composition of HOMO and LUMO. Therefore, the frontier molecular orbital energy levels of **CJ-07** and **CJ-05** are very close, which is consistent with the CV measurements. The calculated energy levels of HOMO and LUMO of each HTM also have the same tendency with regard to the measured results. In the optimal conformation of all HTMs, the terminal triphenylamine group forms a propeller shape, which would reduce the intermolecular aggregation and benefit the amorphous film-forming. On the other hand, the thiophene core and the acetylene-linked benzene ring form an almost coplanar structure, which would promote the π–π interaction between molecules and enhance the charge mobility.

The electrostatic surface potential (ESP) was also simulated for the three HTMs. The positive charge distributes mainly around the hydrogen and nitrogen atoms and the benzene rings, and the negative charge localizes on the high negativity oxygen, sulphur atoms, and acetylene bonds. The dipole moments of **CJ-05**, **CJ-06**, and **CJ-07** were calculated to be 3.46, 2.36, and 0.11 D, respectively, which are all lower than that of spiro-OMeTAD (6.19 D). The low dipole moments of HTMs are unfavourable for the intramolecular charge transfer.

### 2.6. Hole Mobility

As one of the most critical properties of HTMs, the hole mobility of synthesized HTMs and spiro-OMeTAD were measured with space-charge-limited current (SCLC) method. Hole-only devices with the structure of FTO/PEDOT:PSS/pristine HTM/MoO_3_/Ag were fabricated, and their current–voltage responses were recorded in the dark. Linear fit was applied to the SCLC trap-free region based on the Mott–Gurney model [47] (Appendix A), yielding the hole mobility of pristine **CJ-05**, **CJ-06**, and **CJ-07** of 5.56 × 10^−5^, 1.52 × 10^−5^, and 3.91 × 10^−5^ cm^2^ V^−1^ s^−1^, respectively, all of which are inferior to that of spiro-OMeTAD (8.42 × 10^−5^ cm^2^ V^−1^ s^−1^). Therefore, doping seems inevitable during the device fabrication for these HTMs. The acetylene linkers in the three HTMs were supposed to expand the conjugated system, thus promoting intramolecular charge transfer. Meanwhile, the rigid linear structure of acetylene may have altered the patterns of molecular packing, which will definitely affect the charge transfer between molecules. The reliability of the SCLC measurement in this article was also proved by the results of spiro-OMeTAD, which were comparable to those of the reported literature [48].

### 2.7. PL Spectra

In order to understand the hole-transporting dynamics at the HTM-perovskite interfaces in PSCs, we have tested the steady-state PL spectra of the bare perovskite film and pristine HTM-covered perovskite films. It can be seen from Figure 3a that the bare perovskite film gave the strongest emission response, while the introduction of HTM films greatly quenched the fluorescence of perovskite, leaving only 8.12%, 9.45%, 10.27%, and 4.73% of the initial emission intensity for the **CJ-05**-, **CJ-06**-, **CJ-07**-, and spiro-OMeTAD-based samples, respectively (Appendix A and Appendix A). This phenomenon indicates the occurrence of charge separation at the perovskite/HTM interface for all HTMs. The intensity of emission given by the MAPbI_3_/spiro-OMeTAD was the weakest among all samples, implying that the three HTMs all possess weaker hole extraction capability than does spiro-OMeTAD. This result is likely due to the energy levels mismatch between the valence band of perovskite and the HOMOs of HTMs; however, holes could still be injected into them since the energy level mismatch is compensated by the external energy input.

The interfacial hole-transporting dynamics were further investigated by time-resolved PL measurements of the same samples (Figure 3b). The decay curve of each sample was fitted double-exponentially to calculate the related PL lifetime. As listed in Appendix A, the bare perovskite exhibited the longest average lifetime of 121.10 ns among all samples, while the **CJ-05**-, **CJ-06**-, **CJ-07**-, and spiro-OMeTAD-covered perovskite samples presented much shorter lifetimes of 18.17, 19.53, 20.34, and 16.87 ns, respectively. The lifetimes of all samples are consistent with the emission intensity from the steady-state PL measurements.

### 2.8. Hydrophobicity

Since the HTM film not only extracts and transports holes but also protects the underlying perovskite layer from moisture invasion and, thus, enhances the stability of PSCs, the hydrophobicity of HTMs was assessed by measuring the water contact angles with pristine HTM films spin coated on FTO glass. As shown in Appendix A, the water contact angles of **CJ-05**, **CJ-06**, **CJ-07,** and spiro-OMeTAD film are 83.5°, 83.4°, 82.6°, and 83.7°, respectively, which means the three HTMs have similar hydrophobicity to spiro-OMeTAD. The high hydrophobicity is sufficient to construct the moisture-resistant layers for enhancing the device stability.

### 2.9. Film Morphology

Since the HTM film formation is an essential factor that determines the device performance, the surface morphology of doped HTM films deposited on perovskite layers were analysed by scanning electronic microscopy (SEM) and atomic force microscopy (AFM). The dosages of **CJ-05**, **CJ-06**, **CJ-07**, and spiro-OMeTAD were 40.0, 50.0, 50.0, and 72.3 mg/mL, respectively, which were obtained via the optimization of related PSC performance. The dopants applied in the HTMs are lithium bis(trifluoromethylsulfonyl)imide (Li-TFSI), 4-tert-butylpyridine (*t*-BP), and tris(2-(1*H*-pyrazol-1-yl)-4-tert-butylpyridine)-cobalt(III) tris(bis(trifluoromethylsulfonyl)imide)) (FK209). Their dosages are unified with the standard dosages in spiro-OMeTAD solutions (Appendix A). The top view SEM image shows that the pristine MAPbI_3_ layer is tightly assembled with coarse grains (Appendix A), which have the size of approximately 80~450 nm. All the HTM films could completely cover the perovskite layer, which is likely due to the high dosages (Appendix A). However, the pinholes could also be observed on all the HTM films, which may lead to the moisture erosion, thus decreasing the stability of perovskite. It may also cause direct contact between the perovskite layer and the electrode, resulting in a rapid charge recombination. On the **CJ-05** film, the pinholes are sparsely distributed with diameters of approximately 500~660 nm, while the pinholes on **CJ-06** film are dense and small, with diameters of approximately 50~340 nm. The **CJ-07** film presents similar morphology to **CJ-05** film, but the pinholes are slightly denser and larger, with diameters of approximately 700~900 nm. The three HTMs contain less amino groups than spiro-OMeTAD per mass, which may require less dopants, while the actual dosages of the dopants for the three HTMs are the same as spiro-OMeTAD. Thus, the formation of the pinholes may originate from the low miscibility of HTM solutions with Li-TFSI due to the excessive dosages [49]. The film morphology of the three HTMs could likely be improved by optimization of the dopants’ dosages. The films of the three HTMs are generally smooth and flat, except for the pinholes, while the spiro-OMeTAD film is completely rough, with shell-like protrusions and pinholes (with diameters of approximately 100~180 nm). The shell-like protrusions may originate from the aggregation of spiro-OMeTAD molecules induced by the dopants [50].

The AFM images offered a quantitative standard to the morphological comparison. As shown in Appendix A, the root-mean-square (RMS) roughness of MAPbI_3_, **CJ-05**, **CJ-06**, **CJ-07**, and spiro-OMeTAD films were measured to be 15.4, 21.0, 14.7, 29.3, and 36.2 nm, respectively. The depths range of pinholes of **CJ-05**, **CJ-06**, **CJ-07**, and spiro-OMeTAD films were also measured to be approximately 25~120 nm, 15~130 nm, 15~80 nm, and 30~155 nm, respectively. In addition, the height range of protrusion on spiro-OMeTAD film is measured to be approximately 40~125 nm. All the data show that the three HTMs possess better film-forming ability than spiro-OMeTAD.

### 2.10. Device Performance

The n-i-p PSCs with the structure of FTO/compact TiO_2_/mesoporous TiO_2_/MAPbI_3_/HTM/Au were fabricated to investigate the application of **CJ-05**, **CJ-06**, and **CJ-07** on photovoltaic performance as HTM. The detailed fabrication procedure is illustrated in Appendix A. The construction of PSCs was verified by a cross-sectional SEM image (Appendix A) of the **CJ-07**-based device, in which all the component layers could be distinguished clearly. The thicknesses of perovskite and HTM layer could be assessed from the image, which are approximately 400 and 250 nm, respectively. The devices were illuminated under a simulated solar beam calibrated to AM 1.5 G (100 mW cm^−2^) and scanned under reversed voltages, and the response currents were recorded. The optimized dosages of each HTM were extracted from the performances on the basis of different dosages. The current density to voltage (*J*-*V*) curves of the champion cells are shown in Figure 4, and the corresponding photovoltaic parameters are listed in Table 2. The **CJ-05**-, **CJ-06**-, and **CJ-07**-based cells all showed much lower short-circuit current density (*J*_SC_), open-circuit voltages (*V*_OC_), PCE, and fill factor (FF) than spiro-OMeTAD-based cells. This result may originate from the energy level mismatch between the HOMO of the three HTMs and the VB of perovskite; thus, the hole injection from the perovskite was inhibited, and massive charge recombination took place within the perovskite layer. It is noteworthy that even though the three HTMs possess almost the same HOMO levels, their devices showed significantly different *V*_OC_ and PCEs. This phenomenon is likely due to the different morphology of the HTM films. The lower PCEs of **CJ-05**- and **CJ-07**-based cells compared with **CJ-06**-based cells may originate from the increased recombination at the pinhole area, while the increased *V*_OC_ may be ascribed to lower voltage loss across the HTM films. Additionally, series resistance (*R_s_*) of each cell were calculated from the slope near *V*_OC_ of the *J-V* curve, and the order of *R_s_* of all devices is exactly opposite to the order of their *J*_SC_. Due to the poor performance of the devices based on the three HTMS, the stability test was not conducted. Further structural modifications, such as introduction of electron-rich moieties or extension of conjugated system, are supposed to overcome the challenge of energy level mismatch. In addition, the mismatch could also be avoided by applying another perovskite with a lower VB, such as FA- or Cs-containing perovskite, instead of MAPbI_3_.

## 3. Materials and Methods

The details of materials and methods are described in Appendix A.

## 4. Conclusions

In summary, we have realized the syntheses and assessment of three thiophene-based HTMs with acetylene linkers for the PSC application. The introduction of electron-deficient acetylene linkers between the planar thiophene core and twisted triphenylamine groups lowered the energy levels of the frontier molecular orbitals of the HTMs. Meanwhile, its rigid linear structure was also supposed to affect the molecular packing and interactions in a complicated manner. All three HTMs showed better film formation than spiro-OMeTAD. The optimized dosages of the three HTMs for device fabrication were all lower than that of spiro-OMeTAD. However, due to the energy level mismatch, the **CJ-05**-, **CJ-06**-, and **CJ-07**-based cells showed low PCEs of only 6.04%, 6.77%, and 6.48%, respectively. Although the comprehensive effects that acetylene linkers have brought are not satisfactory in this work, the design idea still has much potential to be applied in the design of efficient HTMs.

## Data Availability

The data presented in this study are available in this article.

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
