# Peer review of "Synthesis, Properties, and Application of Small-Molecule Hole-Transporting Materials Based on Acetylene-Linked Thiophene Core"

_molecules, 2023, doi:10.3390/molecules28093739_

Round 1

Reviewer 1 Report

thiophene based hole transport layer with acetylene linkers have been prepared for the application in perovskite solar cells. The properties of the three compounds have been evaluated using optical, thermal, and cyclic voltammetry studies. Further, the hole mobility of the compounds has also been evaluated using hole only devices. PL and TRPL studies claimed the possible exciton quenching and improved charge transport. However, perovskite solar cell performances employing the above prepared materials as HTL yielded poor performances than conventional spiro-OMeTAD due to the energy level mismatch of new HTMs HOMO with perovskite active layer, even though better film formation. This explanation seems be illogical because the difference of energy level mismatch is around only 0.1 eV deeper. The HOMOs are almost same (-5.51, -5.50, -5.52), but the Jsc is different. (8.98, 11.05, 9.00 mAcm-2) Much more critical point would be the pinhole formation or poor interface between perovskite and HTM layer. They need to try the impedance and capacitance analysis. How about the shunt resistance values, indicating leakage currents through the pinholes, even though J-V curve looks normal?

This manuscript should not be considered in the current version, the article needs several modifications.

1.     The reason for reduced solar cell performances need to be discussed in detail.

2.     Why VOC varied with respect to different HTM

3.     In which way, the suggested materials are better than Spiro OMeTAD

4.    Why Tg for CJ-06 not reported, glass transition temperature also important for HTM, why they are lower than Spiro OMeTAD?

5.     From the FESEM images it is clear that the island formation occurs while depositing these three compounds, why island formed, viscosity nature of the solution should be discussed. These islands may contribute reduced performances, more discussion needed.

6.     Possible strategies to improve the currently reported molecule for efficient HTM need to be suggested.

7.     How the intramolecular systems beneficial in HTM applications? (Adv Mater. 2022, 34, 2203794, J.  Mater.  Chem.  A2015, 3, 15024, and org. elec, 2020, 85, 105825)

Reviewer 2 Report

The development of efficient hole-transporting materials for perovskite solar cells is one of the most pressing tasks in the field of organic materials. This work offers a number of new promising compounds for HTMs. An excellent work that not only presents a significant amount of synthetic work, but also assembles model cells and investigates their photophysical properties and efficiency.

The work can be published in its current form.

Author Response

Thanks for your recognition.

Reviewer 3 Report

This paper is about the utilization of new compound for HTL for perovskite solar cells.

However, I cannot see any novelty in this work so it cannot be recommended to be published in Molecules. Moreover there are serious drawback it the discussion part. To make this work suitable for publication authors should clearly demonstrate the potential of the new materials compared to Spiro even at this moment the performance is lower.

1.  First of all it is not discussed the novelty of this work compared to previous publication (Ref. 35). Why does authors chose those compounds for application for HTL? What is the difference from HTL in Ref. 35?

2.      Line 117. “optical properties of CJ-05, CJ-06 and CJ-07 were studied by UV–Vis absorption and photoluminescence (PL) spectra”. How properties can be studied by spectra?

3.  Not only the parameters of champion devices but also statistics on several device of each type should be provided. From the text, it is unclear why the sample Cj-06 demonstrated better performance compared to Cj-05 and -07 even its hole conductivity is lowest. Also performance of sample Cj-05 is lowest even it has high hole conductivity. Considering similarity in the energy structure it is unclear which factors affects sample performance.

4.  How about stability of devices with new HTLs? Is it better compared to devices with Spiro HTLs?

5.      Performance of solar cells was tested only with reverse scans. How about hysteresis? May be devices with new HTLs demonstrated smaller hysteresis compared to those with Spiro HTL?

Round 2

Reviewer 3 Report

The paper can be accepted for publication after authors will implement the data provided in the response letter into the manuscript or SI.